# Oncofertility and Fertility Preservation for Women with Gynecological Malignancies: Where Do We Stand Today?

**DOI:** 10.3390/biom14080943

**Published:** 2024-08-03

**Authors:** Valentina Di Nisio, Nikoletta Daponte, Christina Messini, George Anifandis, Sevastiani Antonouli

**Affiliations:** 1Department of Gynecology and Reproductive Medicine, Karolinska University Hospital, Huddinge, 14186 Stockholm, Sweden; valentina.di.nisio@ki.se; 2Division of Obstetrics and Gynecology, Department of Clinical Science, Intervention and Technology, Karolinska Institutet, Huddinge, 14186 Stockholm, Sweden; 3Department of Obstetrics and Gynaecology, Faculty of Medicine, School of Health Sciences, University of Thessaly, 41500 Larisa, Greece; nikolettadaponte@gmail.com (N.D.); pireaschristina@gmail.com (C.M.); ganif@uth.gr (G.A.)

**Keywords:** oncofertility, gynecological malignancies, cervical cancer, endometrial cancer, ovarian cancer, repro-counseling

## Abstract

Oncofertility is a growing medical and research field that includes two main areas: oncology and reproductive medicine. Nowadays, the percentage of patients surviving cancer has exponentially increased, leading to the need for intervention for fertility preservation in both men and women. Specifically, gynecological malignancies in women pose an additional layer of complexity due to the reproductive organs being affected. In the present review, we report fertility preservation options with a cancer- and stage-specific focus. We explore the drawbacks and the necessity for planning fertility preservation applications during emergency statuses (i.e., the COVID-19 pandemic) and comment on the importance of repro-counseling for multifaceted patients during their oncological and reproductive journey.

## 1. Introduction

The societal and economical transition of the worldwide concept of life, considered as a multifocal interplay of personal lifestyle, social interaction, and career embarkment, leads women to the constant postponement of pregnancy-seeking. Unfortunately, this delay meets with the worrying increase in gynecological malignancies in our era (around 15–20% of the total neoplasms in women), especially in women of childbearing age [1], making them a threat to the patients’ fertility potential in both anatomical (in the case of a necessity for surgical removal of the targeted organ) as well as gonadotoxic anticancer therapies. In this subgroup, around 53% of the cases account for uterine cancer, 25% for ovarian cancer, and less than 15% refer to cervical cancer, while an additional 8% is estimated to be vaginal and vulvar cancer cases [2,3,4,5]. Of note is that approximately 20% of the affected women are younger than the age of 40 and are nulliparous [6,7]. However, the sharp increase in the number of gynecologic oncological cases in young females recorded in the last decades, alongside the strong wish of women for fertility preservation, has enforced the need for the introduction and implementation of conservative and less radical treatment approaches, where possible, in terms for preserving fertility.

The origin of the term “oncofertility” dates back to 2007 when Teresa K. Woodruff described the merge of oncology and reproductive medicine [8]. The broad perspective of the union of these medical disciplines involves patient-tailored anticancer and fertility treatments, including gamete or embryo cryopreservation, together with the pivotal aspect of women’s counseling in psychological, ethical, and legal issues [9]. Among the fertility preservation options available for women diagnosed with early-stage gynecological malignancy, fertility-sparing cytoreductive surgeries remain a feasible approach, while in advanced disease stages, a more radical approach is usually required, leading mainly to infertility. In addition, the use of medical approaches, such as agonists of the gonadotropin-releasing hormone (GnRH), and further assisted reproductive technology (ART) procedures has paved new avenues to the range of possibilities in oncofertility. The aim of the present review is to explore the potential prospects among the available options for fertility preservation on the basis of the best choice depending on the gynecological oncological stage of the disease. Moreover, we focus on putting an accent on the possible difficulties encountered in emergency statuses (e.g., the COVID-19 pandemic) and on the pivotal importance of counseling the patient throughout the medical process and personal recovery.

## 2. Gynecological Malignancies: Stages, Fertility Preservation Feasibility, and Applications

All patients of reproductive age with cancer should be offered oncofertility counseling, regardless of the type and the stage of the disease, in an individualized manner. In cases where fertility preservation has been decided, all patients should be immediately referred to a fertility specialist. Usually, if a patient has been diagnosed with early gynecological cancer at a reproductive age, a fertility sparing approach must be offered and considered. This must be discussed with a team of health professionals who will work together to plan the treatment that is best for the patient and her priorities. This is called a multidisciplinary team (MDT). The MDT looks at national treatment guidelines or the latest evidence for the specific type of cancer.

Fertility preservation options in women include cryopreservation of oocytes and embryos, ovarian tissue, as well as medical and surgical options such as the administration of GnRH agonists and/or ovarian transposition for protection of the ovary against the potentially adverse effects of treatment modalities. Ovarian reserve tests, including either antral follicle count (AFC) or anti-Mullerian hormone (AMH) serum level tests, should be recommended for all individuals in pretreatment for predicting the ovarian response during stimulation and for optimal individualization of the fertility preservation process [10].

Gynecological cancer patients’ survival outcomes have exhibited a significant increase due to the improvement of early detection protocols and advanced treatment. Thus, current attention is focused on the life quality of the surviving patients, like childbearing and fertility potential. The characteristics to be observed when deciding the fertility preservation strategy for oncological patients are related to patient characteristics (e.g., age, physical examination, pelvic ultrasounds, use of partner or donor sperm) and tumor characteristics (e.g., stage, histology appearance, treatment). These characteristics, along with a thorough description of the applicable strategies’ pros and cons, should be discussed during oncofertility counseling with each patient who faces the risk of treatment-induced premature ovarian failure, and this should be followed up in qualified reference centers [11]. During the last years, several reviews have been published in terms of fertility preservation management, mostly focusing on early or late stages of only one type of gynecological malignancy, either cervical [12,13], endometrial [14,15], or ovarian cancer [16,17]. In our present work, we aim to collect the published knowledge and summarize it in a broad and comprehensive perspective under the prism of oncofertility.

Currently, fertility-sparing procedures are offered to young women initially for avoiding invasive and irreversible hysterectomies, salpingectomies, and/or oophorectomies [18]. Among the many fertility preservation options, shielding tissue from radiation damage, conservative gynecological surgeries (the preservation of the uterine corpus or one ovary), fertility-sparing surgical procedures, fertility preservation prior to cytotoxic treatment administration, and ART (as ovarian tissue, oocytes, and embryos cryopreservation) are included [19]. A schematic overview of the fertility preservation approaches in cervical, endometrial, and ovarian cancer is represented in Figure 1.

### 2.1. Cervical Cancer

Cervical cancer mainly affects women of reproductive age. For women wishing for future fertility, conservative treatment approaches are available depending on the stage of cervical cancer:Stage IA1 without lymphovascular space invasion (LVSI) (squamous cell carcinoma, adenocarcinoma, or adenosquamous carcinoma): Conization in a single non-fragmented specimen with free surgical margins (tumor or HSIL) could be performed. In case of positive margins, a cone biopsy should be repeated or a trachelectomy should be performed.Stages IA2–IB1 without LVSI, where the following criteria are met:○Squamous cell carcinoma (any grade) or usual type adenocarcinoma (Grade 1 or 2 only);○Tumor size ≤ 2 cm;○Depth of invasion ≤ 10 mm;○Negative imaging for metastatic disease.Conization with negative ectocervical and endocervical margins and a pelvic lymphadenectomy or sentinel lymph node mapping is recommended. In fact, the sentinel lymph node mapping approach has enabled less radical fertility-sparing operative approaches.Stages IA1 with LVSI, IA2, and IB1 with or without LVSI, where the following criteria are met:○Squamous cell carcinoma, usual type adenocarcinoma or adenosquamous carcinoma;○Tumor size ≤ 2 cm;○Absence of parametrial invasion;○Absence of lymph node metastasis.Radical vaginal trachelectomy should be performed with laparoscopic pelvic lymphadenectomy and cerclage performed at the same surgical time, either vaginally or abdominally [20].

When pelvic radiotherapy is required, ovarian failure occurs. For younger patients, evidence suggests that brachytherapy has a lesser impact when compared to external radiotherapy. For fertility preservation, ovarian transposition is used before radiation to remove the ovaries out of the radiation field.

Most chemotherapeutic drug combinations are known to cause a reduction in the follicle number, leading to premature ovarian failure. Thus, GnRH agonists (GnRHa) are administered, as they provide protection to the ovary by inhibiting the maturation and recruitment of original follicles, thus reducing the toxicity of chemotherapy [21].

According to the American Cancer Society, cervical cancer is the fourth most frequent cancer in women, reporting higher death rates in low- and middle-income countries [22]. The standard treatment for cervical cancer patients is a radical hysterectomy with pelvic lymphadenectomy [23], which includes female infertility as a major irreversible drawback. For fertility preservation in the early stages of cervical cancer, a wide spectrum of conservative treatment modalities has been described, such as cold knife conization (CKC), loop electrosurgical excision procedure/large loop excision of the transformation zone (LEEP/LLETZ), laser conization (LC), as well as trachelectomy [23]. In Stage IA1 cases (micro-invasion, <3 mm), the treatment approach of choice includes CKC, LLETZ, and LC, which can be applied in both micro-invasive squamous cell carcinoma and adenocarcinoma [24]. However, all the above treatment modalities have been linked to an increased risk for Preterm Prelabor Rupture Of Membranes (PPROM) and preterm birth [25]. LLETZ does not affect fertility potential and does not increase the risk of death compared to a hysterectomy [26]. In Stages IA2-IB1, the proposed treatment to spare fertility is a radical vaginal trachelectomy with a pelvic lymphadenectomy when a negative lymph node status is confirmed [27,28]. This procedure (operated using vaginal, laparoscopic, abdominal, and robot-assisted techniques) is regarded as the most established approach for fertility preservation purposes and is applied in both squamous cell carcinoma and adenocarcinoma treatments [28,29,30]. Compared to radical hysterectomy, the literature on the high long-term survival rate (98.4%) and the low relapse rate (4.5%) of this treatment makes this procedure the one of choice in the early stages of cervical cancer [19,31,32]. Future development of non-invasive nuclear methods to identify lymph nodes might improve patient selection for this fertility preservation treatment [33]. The combination of conization and laparoscopic lymphadenectomy is also an option for FIGO IA2-IB1 stage patients with tumors of <20 mm. Although the little data reported in the literature show a 47% success rate in conception and a 97% 5-year disease-free survival rate [34,35], more research on this technique could allow its implementation in the clinical routine for fertility preservation.

Patients with a tumor size of >2 cm have been treated with an initial neoadjuvant chemotherapy followed by combined radical trachelectomy and lymphadenectomy [36,37]. This still-experimental procedure reports high fertility rates and similar oncologic outcomes in comparison with a trachelectomy without additional chemotherapy [38]. When radiotherapy or chemoradiation is required, ovarian transposition is an option to remove the gonads from the radiation field [39,40]. However, the efficacy of this procedure is 50% and dependent on both the radiation dose and scatter [41]. Additionally, the site of transposition is generally difficult to reach during oocyte pick-up for further IVF procedures. The recommendation for these patients is to perform ovarian stimulation for further oocyte/embryo cryopreservation before the initiation of the treatment [42,43]. Nevertheless, potential irradiation-induced damage to the uterus could render the possibility of gestational surrogacy necessary [44]. Whenever oocyte/embryo cryopreservation is not a feasible option, ovarian tissue cryopreservation and subsequent transplantation is a possible choice. Both heterotopic and orthotopic ovarian tissue transplantation can lead to the resumption of ovarian function and, up to today, it brought about the births of 60 babies worldwide [45,46]. The high risk of the reintroduction of cancer cells prompts clinicians to be careful when performing such procedures. Despite all, none of the ovarian tissue cryopreservation cases in women with previous cervical carcinoma resulted in malignancy relapse [42,47]. In cases of radical hysterectomy, the only option for childbearing is womb transplantation [48].

Considering that the primary prevention method for cervical cancer is the vaccination against HPV, it is of interest to briefly introduce the possibility of the vaccination’s side effects on fertility potential. According to the WHO’s Weekly Epidemiological Record of 24 January 2020 regarding the Global Advisory Committee on Vaccine Safety meeting (4–5 December 2019), there is no concrete risk that HPV vaccination could promote premature ovarian insufficiency or a reduction in fertility potential in patients who received the vaccine [49]. In line with this, a recent paper investigating a US cohort of young women (18–33 years old) reported no significant risk of infertility in women vaccinated against HPV for cervical cancer prevention [50].

Regarding the application of ART procedures and infertility rates, a study on patients who had recovered from cervical cancer after a vaginal trachelectomy reported a 13.5% rate of infertility, of which 40% was connected to the cervical factor [51]. The first-line approach for these patients is intrauterine insemination [52].

### 2.2. Endometrial Cancer

Endometrial cancer mainly affects post-menopausal women and only a small percentage may occur in younger women between 20 and 44 years old. On the other hand, hyperplasias are a spectrum of precancerous lesions which generally occur in women of reproductive age.

Fertility-sparing treatment for endometrial cancer does not represent the standard of care; therefore, patient care should be very carefully individualized, and patients should be informed of the need for a hysterectomy immediately after pregnancy or after treatment failure due to the high risk of recurrence. Usually, a bilateral salpingo-oophorectomy is performed, and preservation of the ovaries may be considered in selected pre-menopausal women.

Only patients with atypical hyperplasia or endometrial endometrioid carcinoma which are well differentiated (G1) with no myometrial invasion on imaging, and no genetic risk factors, should be included for fertility-sparing treatment [53].

A hysteroscopically guided endometrial biopsy should be performed in order to confirm the diagnosis. Radiologic staging for the spread of the disease is advised.

Recommended treatment includes oral progestogens (medroxyprogesterone acetate or megestrol acetate) or a levonorgestrel intrauterine device. Patients should be informed in detail that surgical treatment is recommended if no response to progestogens is evidenced after 6 months of therapy [10,54].

The most recent ESGO/ESHRE/ESGE guidance regarding the group of women with endometrial cancer wishing for fertility is based on very careful patient selection, tumor clinicopathological characteristics, treatment, and special issues. Fertility-sparing treatments should be exclusively considered in women with early-stage and non-metastatic disease, and treatment for women with Lynch syndrome should be discussed on a case-by-case basis. The pathological diagnosis of endometrial hyperplasia and endometrial carcinoma is of critical importance for optimal risk stratification and treatment decisions. The G1, G2, G3 grading system is recommended. Fertility-sparing treatment is considered for patients with Grade 1, Stage IA endometrial carcinoma without myometrial invasion and without risk factors. A combined approach consisting of hysteroscopic tumor resection, followed by oral progestins and/or a levonorgestrel intrauterine device, is the most effective fertility-sparing treatment option, although definitive surgery currently represents the recommended approach in cases of non-responders with an inability to conceive, recurrence or disease progression. Additionally, the estrogen and/or progesterone receptor status in each case seems to be predictive of response in conservative treatment decisions; however, could be useful for patient counseling and molecular profiling of early-onset endometrial carcinoma [55].

With more than 400,000 new cases per year, endometrial cancer is the sixth most common cancer in women globally [56]. The risk of developing endometrial carcinoma increases significantly in women affected by obesity and polycystic ovary syndrome [57]. Because of the hormonal sensitivity of this gynecological tumor, the standard approach consists of a hysterectomy and bilateral salpingo-oophorectomy [58].

After myometrial evaluation and the confirmation of the absence of infiltration in both myometrium and extrauterine areas, women diagnosed with endometrial cancer IA can be counseled for fertility-sparing treatments. Progesterone administration, either oral and/or through an intrauterine device (IUD), is the first line of treatment, reaching more than 70% of the positive response rate to the treatment [59,60]. These treatments can be coupled with endometrial curettage to remove any revealed abnormal tissue [61]. In particular, studies report high tumor regression, low relapse, and lower rates of hysterectomy when delivering progesterone through levonorgestrel-releasing IUDs [62,63,64]. This conservative treatment needs follow-up every three months by hysteroscopic examination and endometrial sampling [65]. Moreover, the administration of metformin has shown significant improvements in decreasing the risk of recurrence, even though its use is not included in the current guidelines. Overall, fertility preservation treatment within a combination of progesterone and surgical resection showed 60% of pregnancy outcomes in patients with endometrial cancer IA [66].

Conservative treatment in conjunction with ART procedures should be considered in order to favor a shorter conception time, thus avoiding any drawbacks connected to relapse during pregnancy. Nevertheless, the hormonal stimulation necessary for higher oocyte yield during oocyte pick-up for IVF and further fertilization could have a part in tumor stimulation due to the supraphysiological levels of estrogens. Reports of IVF live births in patients with treated endometrial cancer are promising [67,68,69], taking into account that no increased risk of recurrence rate was outlined [70]. In cases of estrogen-sensitive tumors, either adding letrozole to classical gonadotropin stimulation (i.e., GnRH analogues) or the use of a levonorgestrel IUD has been proposed in endometrial cancer patients to counteract estrogen stimulation on the endometrium [71,72,73]. Once the ART cycles reach the desired purpose, the main option for oncological treatment is to follow the standard of care for endometrial cancer by undergoing a hysterectomy and bilateral salpingo-oophorectomy to reduce the risk of relapse [74].

The last resort for women with unsuccessful ART cycles or advanced stages of endometrial cancer is a gestational surrogacy agreement that allows the birth of a biologically related child without compromising the cancer treatments [75,76]. However, this procedure is allowed only in some countries and involves a third-party woman, namely the surrogate, who will carry and deliver the baby for the receiving couple.

Furthermore, in cases of advanced-stage endometrial cancer requiring urgent hysterectomy, a still-experimental procedure is uterine transplantation from a healthy compatible donor. We will discuss this option further on in the present section.

### 2.3. Ovarian Cancer

Although ovarian cancer usually occurs in post-menopausal women, almost 13% of cases might occur before the age of 40 years and extremely rarely before the age of 19 years. In young patients with early-stage disease, a borderline ovarian tumor (BOT) or a non-epithelial tumor, and a sex cord–stromal tumor at Stage IA or IC, fertility-sparing surgery is acceptable. Depending on histological subtypes and prognostic factors, fertility-sparing surgical planning may differ. Thus, two steps should be followed for making a fertility-sparing decision: firstly, a frozen section should be performed, and the final decision should be taken after the definitive histopathology. Patients with infertility after surgery should be referred to Assisted Reproduction Units with expertise.

BOT:For a serous BOT (Stage IA), unilateral oophorectomy or bilateral cystectomy could be considered depending on patient-specific circumstances.For a serous BOT (Stages IC—III), unilateral oophorectomy or bilateral cystectomy and peritoneal staging, omentectomy, and lymphadenectomy (pelvic and paraortic) could be considered.For a mucinous BOT, unilateral salpingo-oophorectomy could be considered.For a malignant germ cell tumor Grade 1 (G1) unilateral oophorectomy and peritoneal staging, omentectomy and lymphadenectomy (pelvic and paraortic), and adjuvant chemotherapy should be performed.Ovarian Epithelial Cancer:Fertility preservation should be considered only after staging and individualization.For ovarian epithelial cancer Stage IA G1 (serous, mucinous, or endometrioid), unilateral oophorectomy and peritoneal staging, omentectomy, and lymphadenectomy (pelvic and paraortic) should be performed.For ovarian epithelial cancer Stage IA G2–3 (serous, mucinous, or endometrioid), unilateral oophorectomy and peritoneal staging, omentectomy and lymphadenectomy (pelvic and paraortic), and adjuvant chemotherapy should be performed.For ovarian epithelial cancer Stage IC G1, fertility perseveration could be offered using egg donation. More specifically, for ovarian epithelial cancer Stage IC1 or IC2 (low grade), a bilateral salpingo-oophorectomy with uterine preservation could be performed [10].

Fertility-sparing surgery is not recommended in invasive epithelial ovarian cancer cases greater than FIGO Stage I.

Ovarian cancer is the eighth most common cancer occurring in women, reaching a universal estimate of 314,000 cases worldwide [77]. The main ovarian cancer cases where fertility preservation procedures may be recommended are epithelial ovarian cancer, BOTs, and germ cell tumors. Fertility-sparing surgery may be considered for patients who wish to preserve fertility and have an apparent early-stage disease and/or low-risk tumors, such as early-stage invasive epithelial tumors, LMP lesions, malignant germ cell tumors, or malignant sex cord–stromal tumors. Even if the contralateral ovary cannot be spared, uterine preservation can be considered as it allows for potential future assisted reproductive approaches [78]. On the other hand, more advanced or aggressive ovarian cancers (e.g., clear-cell, anaplastic, and small-cell cancers) are excluded from fertility preservation options [79].

The difficulty of the early detection of epithelial ovarian cancers adds more layers of complication and a need to act in a fast and safe manner. The standard of care for these patients includes bilateral salpingo-oophorectomy, hysterectomy, omentectomy, and pelvic and para-aortic lymphadenectomy [80]. If the cancer involves both ovaries, fertility preservation should be discouraged [81,82]. The recommended fertility preservation methods for patients diagnosed with ovarian cancer FIGO Stage IA consist of unilateral salpingo-oophorectomy and staging, omenectomy, and pelvic and para-aortic lymphadenectomy, avoiding sampling the contralateral ovary if macroscopically normal [83]. Patients diagnosed with high-risk early-stage ovarian cancer (IA G3 and up) showed a good response to the applied treatment without impairing 5-year survival rates (ranging from 87 to 95%) [19,83,84]. Furthermore, for patients with high-risk ovarian cancer IAG2 or higher, platinum-based adjuvant chemotherapy can be proposed soon after fertility-sparing surgery [85,86]. In fact, a feasible option for fertility preservation could be the use of ART procedures, after ovarian stimulation, to cryopreserve oocytes and/or embryos. The application of ovarian cortical tissue cryopreservation in these patients is still controversial, because of the high risk of reintroducing malignant cells during the re-transplantation process [87]. In this matter, a possible solution to improve the possibilities of fertility in a safe and risk-free manner for patients could be the use of in vitro artificial ovaries to culture ovarian follicles and obtain mature oocytes for further in vitro fertilization [88]. Albeit still in a fully experimental setting, enormous efforts have been made to establish the most efficient, stabilized, xeno-free, and standardized model of artificial ovaries. More research is needed to implement it for future exploration of its feasible use in clinical settings.

Among epithelial ovarian tumors, BOTs cover up to 20% of cases. In patients with this diagnosis, conservative surgery is highly recommended by adnexectomy of the affected side, or adnexectomy and contralateral cystectomy in cases of bilateral BOTs [89]. A higher recurrence rate is reported in BOT patients undergoing fertility-sparing surgery in comparison with radical treatment [90,91,92,93]. However, studies investigating this procedure in patients with advanced BOT (FIGO IC-III) excluded the association of fertility-sparing surgery with relapse and mortality [94,95]. As for epithelial ovarian cancer and also in cases of non-invasive BOT, oocyte cryopreservation could be a feasible option [96,97]. Individualized treatment after discussion in MDT meetings must be offered where different approaches for serous BOTs and mucinous BOTs are usually considered.

Germ cell tumors, even if rare, affect young women, thus making necessary the possibility of undergoing fertility-sparing surgeries. The standard treatment consists of unilateral adnexectomy, peritoneal staging, and omentectomy, possibly followed by adjuvant chemotherapy with bleomycin, etoposide, and cisplatin [98,99]. Interestingly, these treatments report high reproductive outcomes without affecting the risk of teratogenicity [100]. Fertility sparing is highly recommended in cases of yolk-sac tumors [101] and pure dysgerminoma [102], and it should be carefully followed up in G2–3 and advanced-stage immature teratoma [103,104]. In particular, the administration of adjuvant chemotherapy in cases of high-stage yolk-sac tumors, early-stage dysgerminoma with relapse, and immature ovarian teratoma Stage I G2–3 is highly recommended, due to higher overall survival rates and unaffected fertility rates [105,106,107,108]. High survival and fertility rates, together with low relapse rates give to these patients the chance to achieve pregnancy without the need to undergo ART cycles [109].

Interestingly, there is an increased risk for patients carrying *BRCA* mutations to report a reduced ovarian pool and to develop OC, due to the disruption of the DNA repair mechanisms [110]. Particularly for those harboring germline mutation of *BRCA1/2*, the often-proposed fertility preservation option is the risk-reducing surgery of bilateral salpingo-oophorectomy before the age of 40 [111].

Overall, due to the high gain–high risk of fertility-sparing procedures in gynecological cancer patients, it is fundamental to carry out such treatments following international guidelines and clinical/ethical best practices [20,112]. Each decision should be accompanied also by a comprehensive individual patient assessment, taking into consideration any genetic risk factors [113].

### 2.4. Fertility Cryopreservation Techniques

In the multifaceted reality of ART, oocyte and embryo cryopreservation is the most well established and standardized procedure for assisted fertility preservation. Particularly, oncological patients should undergo ovarian stimulation and oocyte/embryo cryopreservation before undergoing cancer therapy.

Patients who do not have an urgency to start cancer therapy and who either have a partner or wish to use donor sperm can be counseled for embryo cryopreservation. Whenever feasible, this should be considered as the preferred option, because of higher pregnancy rates in comparison to oocyte cryopreservation [114,115]. Both oocyte and embryo cryopreservation require one cycle of gonadotropin-based ovarian stimulation of the woman, after which the stimulated oocytes will be retrieved through an ultrasound-guided procedure under local sedation. Afterward, the picked oocytes can undergo immediate cryopreservation (if the patient does not have a partner and refuses the use of donated sperm), or in vitro fertilization to obtain the embryo that will be cryopreserved [114,116].

Another feasible option is ovarian tissue cryopreservation. The procedure is not yet highly standardized, and a few studies report ovarian function recovery in 2–8 months and live babies born following autotransplantation of the ovarian cortical biopsy after anticancer treatment [117,118,119,120]. The guidelines directing the procedures and the feasibility of ovarian tissue cryopreservation are still controversial depending on the country and its internal legislation. In fact, in some countries ovarian tissue cryopreservation and re-transplantation is still considered as an experimental approach and should be carefully considered before proposing it to patients. Notably, in prepubertal girls and patients who cannot delay the start of chemotherapy or radiotherapy, this is the only option for fertility preservation [121]. One of the main disadvantages of this procedure concerns the risk of reintroducing malignant cells during the re-transplantation of the ovarian biopsy, especially in patients diagnosed with ovarian cancer or leukemia [122]. Also, a comprehensive meta-analysis on early-stage cervical adenocarcinoma and the safety of the ovarian cryopreservation procedure reported an increased incidence in non-squamous type cancers, concluding that the histotype and stage evaluation of cervical cancer should be considered to allow the choice of a highly personalized fertility preservation method during oncofertility counseling [123].

### 2.5. Other Fertility Preservation Techniques

Radiotherapy is an anticancer treatment that aims at damaging highly proliferative malignant cells. The area subjected to the radiation is fully involved, causing ionizing damage also in normal tissue. This untargeted treatment may provoke damage to follicles, due to the gonadotoxic effect of radiation [123]. In order to protect the ovary from the detrimental effect of radiation, patients can be counseled for ovarian transposition. Briefly, this procedure consists of cutting the utero-ovarian ligaments and moving the ovaries from the original place to a safer one less exposed to the radiation field [124]. Depending on the targeted area, the ovaries will be fixed in different areas. For instance, the gonads will be moved laterally for craniospinal irradiation, while they need to be moved outside the pelvis before pelvic irradiation [125] or into the neighboring extraperitoneal area [126]. Among the risks of the ovarian transposition, postoperative adhesion, chronic pelvic pain, migration to the native position, increased ovarian cyst formation, and premature ovarian failure can be experienced [127]. Only 1% of patients could encounter the risk of metastasis in the ovaries [128]. Due to the different location of the ovaries, standard transvaginal oocyte recovery is not feasible during IVF cycles, thus requiring more invasive transabdominal oocyte retrieval [118]. In cervical cancer patients, the success rate of ovarian transposition is approximately 90%, making it a procedure of choice whenever possible [127]. However, in uterine cervical cancer, a combined approach of ovarian transposition and cryopreservation could increase the chances of fertility restoration [39]. In the end, ovarian transposition should not be recommended to patients who need to undergo both chemo- and radiotherapy since the administration of gonadotoxic chemotherapeutic drugs is required.

A potential fertility preservation treatment is the use of GnRHa. This compound is frequently used for ovarian stimulation protocols, and its main role is the suppression of ovarian function by reducing gonadotropin levels [129]. The main effect of GnRHa administration is to induce a “flare-up” effect in the ovarian microenvironment that will decrease after two weeks due to the downregulation of GnRH receptors [130]. GnRHa administration during chemotherapy should be administered 2–4 weeks before starting anticancer treatment and it could help in reducing ovarian damage, even if the results are still conflicting [131]. Therefore, this procedure should be considered for urgent cases in the conservative treatment of ovarian cancer, or together with neoadjuvant administration before conservative procedures in early-stage cervical cancer patients [132,133]. Nevertheless, GnRHa protective administration is still controversial, depending on the patient’s age and the nature of the chemotherapy treatment [134]. In fact, while GnRHa co-treatment is an established procedure for ovarian protection and fertility preservation in patients diagnosed with breast cancer and hematological malignancies, it still needs further clinical research to be standardized in gynecological cancers. Moreover, the risk of inducing ovarian hyperstimulation syndrome adds another layer of uncertainty in the application of this co-treatment. A possible solution to this drawback could be the use of GnRH antagonists that suppress gonadotropin secretion, whilst avoiding the “flare-up” effect and reducing the risk of related ovarian hyperstimulation syndrome [135]. Overall, studies on patients diagnosed with gynecological cancers and undergoing ovarian stimulation for ART treatments reported no significant increased risk of cancer recurrence in patients with endometrial cancer (up to 38% vs. 56% in stimulated and unstimulated groups, respectively [74,136]), with the same result achieved in a cohort of ninety patients diagnosed with gynecological cancers (among which were ovarian, endometrial, uterine, and cervical; a total recurrence rate of 6% vs. 9% in stimulated and unstimulated patients, respectively [137]).

Uterine transplantation is an experimental surgical procedure to be considered in the advanced stages of cancers, and when fertility preservation through conservative treatments is not a feasible option. In the literature, few attempts have been recorded for uterine transplantation, following extensive counseling and patient-specific evaluation [138]. The process demands the patient to undergo ovarian stimulation and oocyte retrieval for IVF procedures, followed by embryo cryopreservation. Afterward, the uterine vasculature needs to be carefully preserved during the donor’s hysterectomy, pivotal for a successful transplant in the receiving patient [139]. One year before transplant and during the whole pregnancy, the patients need to undergo immunosuppressive protocols (e.g., mycophenolate mofetil, methylprednisolone, antithymocyte antibody, and thymoglobulin, tacrolimus, and prednisolone treatment) to avoid graft rejection and graft-versus-host disease [138,140]. The use of immunosuppressors could be a concern in virus-derived cervical cancer patients, who may have recurrent cancer due to this treatment [141]. The first trial with a successful pregnancy, delivery, and live birth was reported in 2014 by Brännström and collaborators [48]. However, several trials describe the development of uterine intravascular thrombosis after organ transplantation as a main risk [142]. Since there is a lack of published results, this option is still considered as experimental; nevertheless, the American Society for Reproductive Medicine is actively developing multidisciplinary uterine transplant guidelines [143].

Taking everything into account from this section, fertility preservation options for patients diagnosed with diverse stages of gynecological malignancies are described and summarized in Table 1. The importance of these procedures relies on the improvement of the health and the mental and social well-being of patients. Despite this, careful considerations need to be undertaken regarding the cancer staging and the feasibility of the procedure in a patient-specific fashion. The cryopreservation of embryos and oocytes is still considered the standard-of-care option. In addition, the improvement of medical and surgical procedures should be regulated in a systematic way, allowing the desired maternity for patients that cannot follow the regular ART cycle. For this, specialized multidisciplinary centers and teams (i.e., gynecologists, radiation and medical oncologists, reproductive endocrinologists, perinatologists, and psychologists) should counsel and evaluate the ongoing treatment of the patient, well defining the risk of disease recurrence.

## 3. Current Considerations in Oncofertility

### 3.1. The Impact of the COVID-19 Pandemic

In 2020, the whole world found itself facing a new invisible enemy, namely the new coronavirus COVID-19, finally declared a pandemic by the World Health Organization in March 2020 [146]. The severe acute respiratory syndrome coronavirus 2 (SARS-CoV-2) affected young and old people, causing life-threatening complications, particularly in patients with ongoing diseases or comorbidities [147]. The emergency state caused a great impact on hospital manageability and patient care, with unfortunate repercussions on oncological patients, who displayed a 2.3 times higher risk of getting infected by SARS-CoV-2 with a higher mortality rate [148,149,150]. For instance, in the Netherlands, the surgical care of patients with cervical cancers dropped, as well as the administration of non-surgical treatments for advanced-stage ovarian cancer [151]. Additionally, the high limitation in starting fertility preservation procedures for safety reasons before gonadotoxic anticancer therapy administration created a mental, social, and personal burden on patients who were not able to undergo either surgical- or ART-related fertility preservation, who then gave up on the desire to procreate [152,153,154].

Despite the limitations caused by the highly infectious nature of SARS-CoV-2, the side effects, such as the closure of fertility units, postponements of surgeries, delays in treatments and hormonal cycles, and a reduction in medical and interdisciplinary consultations, hindered the application of processes for oncofertility purposes [155]. In the worst cases, fertility preservation programs were interrupted where the pandemic caused high mortality [156]. Depending on the country and on the relative weight that the COVID-19 pandemic had on it, the non-emergency nature of the treatment, as well as the shortage in hospital staff and the increased risk in crowded closed areas, lead oncologists to conduct continuous referrals of these procedures [157]. Broadly, due to the SARS-CoV-2 vulnerability of cancer patients because of immunosuppression, the visits from fertility centers were conducted in a webinar manner, allowing safety during the consultation for the couple. Whenever it was imperative that the medical visit required the patient to access the clinic, strict rules on the appointment basis, banned visitors, a safety personal distance of 2 m apart, and plexiglass dividers were applied, together with a thorough disinfection of the room and the common area between patients [158].

With the restriction on the staff present in fertility centers or hospitals, the outbreak of the COVID-19 pandemic also influenced the concomitant presence of interdisciplinary experts who should constantly follow and counsel the patient during the fertility preservation process [159]. In fact, several fertility clinics from diverse countries suffered the impact of a missing psychologist figure in the overall picture, resulting in an impaired support system, ineffective consultation, and lower requests for fertility preservation from cancer patients [159,160,161,162].

To this end, important lessons can be extrapolated from these troublesome times, mainly underlining the utmost importance of gynecologic oncologists to ensure the continuum of health care of their patients, nowadays and in any future health emergency situations. Indeed, some European countries started improving the system by prioritizing high-risk cancer women, implementing video-supported consultations for counseling, prescription of medications, and long-term treatment management, while also increasing screening awareness campaigns [163]. On the medical staff side, the support from gynecological oncology trainees could release stress levels, along with the anxiety, depression, and low professional fulfillment that affected medical staff worldwide during the COVID-19 pandemic [164,165]. Nevertheless, the fight against an emergency such as the COVID-19 infectious disease should be supported not only by single clinics or hospitals, but also by efforts from local, national, and global entities evaluating and helping the necessities of the medical and societal environment [166].

A further consideration to bear in mind that has been a clear side effect of the COVID-19 pandemic is the possible side effects of administered COVID-19 vaccines on male and female fertility potential [167]. In fact, a fluctuation in sperm parameters of men who received the vaccination has been reported, even though this side effect does not drastically influence the possibility of pregnancy [168,169,170]. On the other hand, besides irregular menstruation and vaginal bleeding [171,172], no relevant side effects were revealed in vaccinated women on pregnancy outcomes. Albeit the presence of the SARS-CoV-2 mediator Angiotensin-converting enzyme 2 (ACE2) in several tissues of the female reproductive system and in oocytes, its relatively low concentration does not seem to cause viral infection of female reproductive tissues and cells [173]. Nevertheless, a follow-up clinical study investigating the reproductive outcomes of oncological patients with and without vaccination is not present in the literature, leaving the question of COVID-19 vaccination and oncofertility still an open door for further studies.

### 3.2. The Significance of Repro-Counseling and Managing in Cancer Patients for Fertility Preservation

The emergence of oncofertility as a multifaceted and interdisciplinary medical approach to the combination of cancer and fertility treatments includes, besides medical and technical management, the important branch of psychological support of the patient undergoing such treatments and the whole curative and childbearing procedure. Today’s advancements in surgical and medical procedures have enabled patients with what was considered “irreversible infertility” to have biological and genetically related newborns, thanks to groundbreaking methodologies such as uterine transplantation and tissue bioengineering [174,175]. In the process of granting awareness to patients of what are the case-specific possibilities, other figures than the oncologist, who in one-third of the cases is not able to explain fertility issues, have a crucial role, such as oncology nurses, mental health professionals, and generic counselors [23,118].

The impact over the long term on the psychology and quality of life of patients after cancer and fertility treatments should be addressed. Among the main outcomes of the deterioration in quality of life are depression and anxiety disorders [176,177]. The first step of this two-way patient–psychologist process, from now on referred to as “repro-counseling”, is identifying women with higher risks of developing psychological and sexual disorders, as assessed by the Brief Index of Sexual Functioning for Women [178] or the Female Sexual Function Index [179]. Few studies found a positive association between successful repro-counseling and an increased quality of life in cancer survivors [180,181,182]. Despite the impact of financial resources, social and family prejudices, or a lack of support, a serious downside in repro-counseling is the specific communication strategies needed to explain the available options of fertility preservation and their efficacy [183]. In some cases, decision aids could reduce biases by clarifying the advantages and disadvantages of each fertility preservation process [9,184], thus helping the patients take one step towards thoughtful decision-making.

## 4. Upcoming Future in Oncofertility and Concluding Remarks

The pre-clinical and clinical advancements performed in the field of oncofertility are nowadays based on the strong research outputs of hospitals and IVF laboratories worldwide. However, the path to consolidated standards for all gynecological cancer cases is still far from reality. New, improved protocols and technological advancements are still to be fully understood and developed to allow the majority of women to fulfill their desire for motherhood. For instance, the development of 3D bioengineered ovaries has been accomplished in mice, resulting in both hormonal and fertility restoration in ovariectomized animals [185]. These achievements lead the way to a translational medicine perspective, also giving hope to women affected by advanced-stage cancer and thus needing radical surgical excision of the affected organs. The future of fertility preservation resides not only in the engineering of organs but also in the genomic investigation of potential fertility markers and/or sensitivity to chemotherapy. This is also the case for the optimization of in vitro follicle activation and oocyte maturation protocols, along with the implementation of protection for patients undergoing gonadotoxic treatment (i.e., “fertoprotective” agents).

Furthermore, deeper research in patient selection and long-term oncological and fertility follow-ups is necessary to enlarge the body of proof present in the literature, giving air to the possibility of developing personalized therapies targeted mostly at cancer patients’ needs [186]. Initiating the process from the effectiveness of surgical fertility-sparing procedures, continuing with golden standard ART-based fertility preservation through embryo or oocyte cryopreservation, and accompanied by extensive specialized patient and family counseling for fertility preservation, the combined work of physicians and patients can eventually restore hope and give new strength to these women.

## Figures and Tables

**Figure 1 biomolecules-14-00943-f001:**
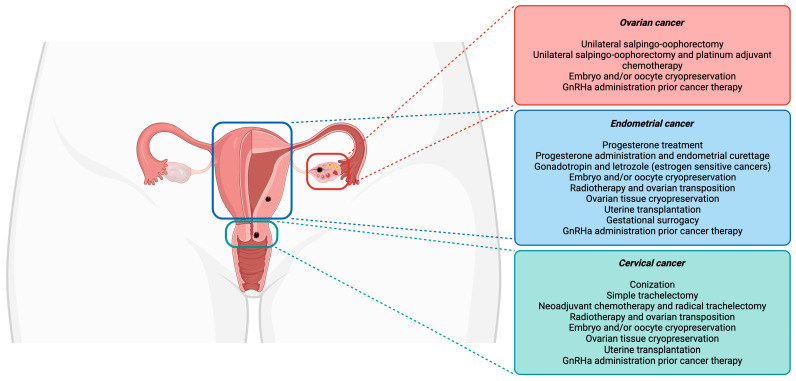
Fertility preservation approaches in gynecological cancers.

**Table 1 biomolecules-14-00943-t001:** Gynecological cancers, staging, and fertility preservation applications.

Cancer	Stage/Type	Fertility Preservation Treatment	Reproductive Outcome	Reference
Cervical cancer	FIGO Stage IA1	Large loop excision of the transformation zone	No effect on fertility potential	[24,26]
	FIGO Stages IA2-IB1	Simple trachelectomy with pelvic lymphadenectomy	High long-term survival rate and low relapse rate No effect on fertility potential	[19,31,32]
		Conization and laparoscopic lymphadenectomy	Good conception rates and high 5-year disease-free survival rate	[34,35]
	FIGO Stages IB1 and up	Neoadjuvant chemotherapy, radical trachelectomy, and lymphadenectomy	High fertility rates and low recurrence rates	[38]
	FIGO Stages IB1-IIB (patients ≤ 40 y.o.)	Radiotherapy/chemoradiation and ovarian transposition	Good efficacy and high preservation of ovarian function	[39,40]
		Radiotherapy and ovarian tissue cryopreservation	No pregnancies have been achieved yet	[144]
	FIGO Stage II onwards	Radical hysterectomy, uterine transplantation, or gestational surrogacy	Low risk of cancer spreading outside the confined area Uterine transplantation is still experimental and needs more literature data	[48]
	Any stage	Ovarian stimulation before treatment followed by oocyte/embryo cryopreservation	High pregnancy rates and live birth rates Higher risk of recurrence in patients with FIGO stage IB and on	[51]
Endometrial cancer	FIGO Stage IA	Progesterone administration (oral and/or intrauterine device)	70% positive response rate Almost 50% pregnancy outcome, of which 70% reach live birth	[60,67]
		Progesterone administration and surgical resection	Up to 60% pregnancy outcome No increased recurrence rate	[66,67]
		Progesterone administration and/or GnRH agonist (GnRHa) treatment	Almost 100% complete response More than 50% pregnancy outcome and 20% live birth rates	[67]
	FIGO Stage I	Progesterone administration, GnRHa treatment, and endometrial curettage IVF procedure	Higher rates compared to spontaneous conception Need to follow up for recurrence risk	[68,69]
		Progesterone administration, and/or GnRHa treatment IVF/ICSI procedure	50% live birth rates, ca. 25% of recurrence Hysterectomy and bilateral salpingo-oophorectomy after pregnancy	[74]
	FIGO Stage II onwards	Gestational surrogacy	-	[75,76]
Epithelial ovarian cancer	FIGO Stages IA—IA G3	Unilateral salpingo-oophorectomy, omentectomy, and pelvic and para-aortic lymphadenectomy	Lower recurrence rate compared to FIGO stages IC Higher survival rates Preservation of reproductive and endocrine functions	[19,83,84]
	FIGO Stage IA G2 onwards	Unilateral salpingo-oophorectomy, omentectomy, pelvic and para-aortic lymphadenectomy, and platinum-based adjuvant chemotherapy	Conception rate ranging between 60 and 100%	[85,86]
Borderline ovarian tumor	FIGO Stage I	Adnexectomy of the affected side/adnexectomy and contralateral cystectomy (bilateral tumors)	Low recurrence rate and more than 50% pregnancy rates	[89]
	FIGO Stages IC—III	Adnexectomy of the affected side/adnexectomy and contralateral cystectomy (bilateral tumors)	Relapse rate up to 45% Pregnancy rates higher than 80%	[94]
Germ cells tumors	All stages	Unilateral adnexectomy and omentectomy, followed by adjuvant chemotherapy (bleomycin–etoposide–cisplatin)	High reproductive outcomes, no teratogenicity risk	[100]
	Yolk-sac tumors	Laparotomy and tumor reduction and oocyte cryopreservation, followed by adjuvant chemotherapy (bleomycin–etoposide–cisplatin)	High 5-year survival rates (90%), low recurrence, and high pregnancy rates	[101,102]
	Dysgerminoma	Unilateral salpingo-oophorectomy, followed by adjuvant chemotherapy (bleomycin–etoposide–cisplatin) in advanced stages	High long-term survival rates (up to 100%) Pregnancy rates up to 45%	[102]
	Immature teratoma	Unilateral salpingo-oophorectomy and complete staging, followed by adjuvant chemotherapy (bleomycin–etoposide–cisplatin) in advanced stages (>IA G1)	Overall survival up to 90% Pregnancy rates more than 60%	[104]
Ovarian cancer (general)	Any stage	Ovarian stimulation before treatment followed by oocyte/embryo cryopreservation	Pregnancy rates up to 12% after oocyte cryopreservation and up to 40% after embryo cryopreservation	[145]
		Ovarian tissue cryopreservation	High risk of reintroduction of malignant cells	[87]

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
