# Peer review of "Oncofertility and Fertility Preservation for Women with Gynecological Malignancies: Where Do We Stand Today?"

_biomolecules, 2024, doi:10.3390/biom14080943_

Round 1

Reviewer 1 Report

Comments and Suggestions for Authors

In the manuscript entitled “Oncofertility and fertility preservation for women with gynecological malignancies: where do we stand today?” ,the authors present data from a review of the literature to explore the potential for directing stage-specific selection among options for fertility preservation.

This is an interesting data in the field of management of gynecological cancer patients who desire fertility preservation. However, because of the wide variety of content, I have the impression that each has not been analyzed in depth. Some comments are listed below.

2.1.Cervical cancer

:

1)    The auther mentioned “For that when GnRH agonist is administered, it provides protection to the ovary by inhibiting the maturation and recruitment of original follicles, thus reducing the toxicity of chemotherapy”. However, the use of GnRH agonists is controversial, as efficacy is sparse depending on the age of the patient and the nature of the chemotherapy (Int J Mol sci. 2022).  

2)    Regarding Table 1, treatment and outcome are misaligned and difficult to read.

Author Response

In the manuscript entitled “Oncofertility and fertility preservation for women with gynecological malignancies: where do we stand today?” ,the authors present data from a review of the literature to explore the potential for directing stage-specific selection among options for fertility preservation.

This is an interesting data in the field of management of gynecological cancer patients who desire fertility preservation. However, because of the wide variety of content, I have the impression that each has not been analyzed in depth. Some comments are listed below.

2.1.Cervical cancer

1)    The auther mentioned “For that when GnRH agonist is administered, it provides protection to the ovary by inhibiting the maturation and recruitment of original follicles, thus reducing the toxicity of chemotherapy”. However, the use of GnRH agonists is controversial, as efficacy is sparse depending on the age of the patient and the nature of the chemotherapy (Int J Mol sci. 2022). 

Answer: Thank you for your useful suggestion. We modified the layout of the review by merging parts 2 and 3 in one whole and more comprehensive section of the review “2. Gynecological malignancies: stages and fertility preservation feasibility and applications”, as requested by another reviewer. We agree with your comment and therefore, we added this information and expanded on the topic in the new section (lines 1017-1048):

“Nevertheless, the GnRH agonist protective administration is still controversial, de-pending on patient’s age and nature of the chemotherapy treatment [135]. In fact, while the GnRHa co-treatment is an established procedure for ovarian protection and fertility preservation of patients diagnosed with breast cancer and hematological malignancies, it still needs further clinical research to be standardized in gynecological cancers. Moreover, the risk of inducing ovarian hyperstimulation syndrome adds another layer of uncer-tainties in the application of this co-treatment. A possible solution to this drawback could be the use of GnRH antagonists that suppress the gonadotropins secretion, whilst avoiding the “flare-up” effect and reducing the risk of related ovarian hyperstimulation syndrome [136].”

2)    Regarding Table 1, treatment and outcome are misaligned and difficult to read.

Answer: Thank you for your comment. We modified the layout of the table to make it clearer and more readable.

Reviewer 2 Report

Comments and Suggestions for Authors

In this manuscript, entitled “Oncofertility and fertility preservation for women with gynecological malignancies:  where do we stand today?” the authors outline the fertility preservation options in the context of the specific cancer type and stage. In addition, they recall the importance of the counseling regarding reproductive plans for young patients facing cancer therapies.

The topic is very interesting, however, in the present form the manuscript is not suitable for publication in Biomolecules.

With major revisions, the manuscript will find its place in Biomolecules

The review requires a clearer re-organization, maybe more schematic, of different types of gynecological malignancies and stages together with the fertility preservation options.

Indeed, in the manuscript, two Sections entitled “Cervical cancer” are present either in line 66 and in line 173, the same for section entitled “endometrial cancer” see line 100 and line 222 respectively, and for section entitled “ovarian cancer” line 119 and line 262.

The authors should check carefully the references cited in the text, for example line 499 the reference [157] is not correct. Another example, line 510 the authors wrote [158] however, the reference numbered 158 is missing in the References section.

It is desirable that the authors cite more recent papers; several reviews about this topic have been published and it may be of interest to discuss them to sum up the state of the art.

This reviewer may suggest the authors to consider a short paragraph about fertility preservation option for patients carrying BRCA1 and BRCA2 mutations in the germline.

Minor comments

- Readability of the table 1 can be improved

Comments on the Quality of English Language

minor comments

- Writing of the manuscript can be improved

Author Response

In this manuscript, entitled “Oncofertility and fertility preservation for women with gynecological malignancies:  where do we stand today?” the authors outline the fertility preservation options in the context of the specific cancer type and stage. In addition, they recall the importance of the counseling regarding reproductive plans for young patients facing cancer therapies.

The topic is very interesting, however, in the present form the manuscript is not suitable for publication in Biomolecules.

With major revisions, the manuscript will find its place in Biomolecules

The review requires a clearer re-organization, maybe more schematic, of different types of gynecological malignancies and stages together with the fertility preservation options.

Indeed, in the manuscript, two Sections entitled “Cervical cancer” are present either in line 66 and in line 173, the same for section entitled “endometrial cancer” see line 100 and line 222 respectively, and for section entitled “ovarian cancer” line 119 and line 262.

Answer: Thank you for your useful suggestion. On your request, we modified the layout of the review by merging parts 2 and 3 in one whole and more comprehensive section of the review: “2. Gynecological malignancies: stages and fertility preservation feasibility and applications”.

The authors should check carefully the references cited in the text, for example line 499 the reference [157] is not correct. Another example, line 510 the authors wrote [158] however, the reference numbered 158 is missing in the References section.

Answer: Thank you for your comment. We checked all the references after the manuscript applied revision.

It is desirable that the authors cite more recent papers; several reviews about this topic have been published and it may be of interest to discuss them to sum up the state of the art.

Answer: Thank you for your suggestion. We included a small paragraph introducing the state of art and acknowledging the recent reviews published in the topic (lines 232-237):

“During the last years, several reviews have been published in terms of fertility preservation management, mostly focusing on early or late stages of only one type of gynecological malignancy, either cervical [12,13], endometrial [14,15], or ovarian cancer [16,17]. In our present work, we aim at collecting the published knowledge and sum-marize it in a broad and comprehensive perspective under the prism of oncofertility.”

This reviewer may suggest the authors to consider a short paragraph about fertility preservation option for patients carrying BRCA1 and BRCA2 mutations in the germline.

Answer: Thank you for your suggestion. We included a small paragraph (lines 798-802) filling the missing information, following your request.

“Interestingly, there is an increased risk for patients carrying BRCA mutations to report a reduced ovarian pool and to develop OC, due to the disruption of the DNA repair mechanisms [110]. Particularly for those harboring germline mutation of BRCA1/2 the often-proposed fertility preservation option is the risk-reducing surgery of bilateral sal-pingo-oophorectomy before the age of 40 [111].”

Minor comments

- Readability of the table 1 can be improved

Answer: Thank you for your comment. We modified the layout of the table to make it clearer and more readable.

Reviewer 3 Report

Comments and Suggestions for Authors

The manuscript contributes to giving information about women with cancer and how to preserve their oncofertility and fertility. The authors described well the information based on reported data including recent studies. I believe that the accumulation of information such as this manuscript is very important for future clinical applications.

Major points

1. It would be better to add examples of side effects of vaccinations and points for improvement in each section (For example, GnRH agonist and so on.).

2. If there are any preventive methods for women who have not yet with cancer, please add them.

Minor points

1. Line 365: Check againovarian transposition is ca.”. Is it correct?

Comments on the Quality of English Language

Minor editing of English language required.

Author Response

The manuscript contributes to giving information about women with cancer and how to preserve their oncofertility and fertility. The authors described well the information based on reported data including recent studies. I believe that the accumulation of information such as this manuscript is very important for future clinical applications.

Major points

  1. It would be better to add examples of side effects of vaccinations and points for improvement in each section (For example, GnRH agonist and so on.).

Answer: Thank you for your suggestion. We now included in the revised manuscript a part acknowledging side effects of:

  • HPV vaccination (lines 543-551):

“Considering that the primary prevention method for cervical cancer is the vaccination against HPV, it is of interest to briefly introduce the possibility of vaccination’s side effects on the fertility potential. According to the WHO Weekly Epidemiological Record of 24 January 2020 of the Global Advisory Committee on Vaccine Safety meeting (4-5 December 2019), there is no concrete risk that HPV vaccination could promote premature ovarian insufficiency or reduction of fertility potential in patients who received the vaccine [49]. In line with this, a recent paper investigating a US cohort of young women (18-33 years old) reported no significant risk of infertility in women vaccinated against HPV for cervical cancer prevention [50].”

  • Covid-19 vaccination (lines 1188-1200):

“A further consideration to bear in mind that has been a clear side effect of Covid-19 pandemic, is the possible side effects of administered Covid-19 vaccines to male and female fertility potential [169]. In fact, it has been reported a fluctuation in sperm pa-rameters of men that received the vaccination, even though this side effect does not drastically influence the possibility of pregnancy [170–172]. On the other hand, beside irregular menstruation and vaginal bleeding [173,174], no relevant side effects were revealed in vaccinated women on pregnancy outcomes. Albeit the presence of SARS-CoV-2 mediator Angiotensin-converting enzyme 2 (ACE2) in several tissues of the female reproductive system and in oocytes, its relatively low concentration seems to be unlikely to be used for viral infection of female reproductive tissues and cells [175]. Nevertheless, in literature is not present a follow-up clinical study investigating the re-productive outcomes of oncological patients with and without vaccination, leaving the question of Covid-19 vaccination and oncofertility still an open door for further studies.”

Furthermore, we included and expanded some points of improvements:

  • GnRH agonist/antagonist use (lines 1017-1048):

“Nevertheless, the GnRH agonist protective administration is still controversial, de-pending on patient’s age and nature of the chemotherapy treatment [135]. In fact, while the GnRHa co-treatment is an established procedure for ovarian protection and fertility preservation of patients diagnosed with breast cancer and hematological malignancies, it still needs further clinical research to be standardized in gynecological cancers. Moreover, the risk of inducing ovarian hyperstimulation syndrome adds another layer of uncer-tainties in the application of this co-treatment. A possible solution to this drawback could be the use of GnRH antagonists that suppress the gonadotropins secretion, whilst avoiding the “flare-up” effect and reducing the risk of related ovarian hyperstimulation syndrome [136].”

  • Artificial ovary for OC patients (lines 745-750):

“In this matter, a possible solution to improve the possibilities of fertility in a safe and risk-free manner for the patients could be the use of in vitro artificial ovaries to culture ovarian follicles and obtain mature oocytes for further in vitro fertilization [88]. Albeit still in a fully experimental setting, enormous efforts have been done to establish the most efficient, stabilized, xeno-free, and standardized model of artificial ovary. More research is needed to implement it for future exploration of the feasible use in clinical settings.”

  • Uterine transplantation (lines 667-669, 1054-1072):

“Furthermore, in cases of advanced stage endometrial cancer requiring urgent hysterectomy, a still experimental procedure is the uterine transplantation from a healthy compatible donor. We will discuss this option further on in the present section.”

“Uterine transplantation is an experimental surgical procedure to be considered in advanced stages of cancers, and when fertility preservation through conservative treatments is not a feasible option. In literature, few attempts have been recorded for uterine transplant, following extensive counseling and patient-specific evaluation [140]. The process demands the patient to undergo ovarian stimulation and oocyte retrieval for IVF procedures, followed by embryo cryopreservation. Afterwards, the donor hyster-ectomy needs to carefully preserve the uterine vasculature, pivotal for a successful transplant in the receiving patient [141]. One year before transplant and during the whole pregnancy, the patients necessitate to undergo immunosuppressive protocols (e.g., my-cophenolate mofetil, methylprednisolone, antithymocyte antibody and thymoglobulin, tacrolimus, and prednisolone) to avoid graft rejection and graft-versus-host disease [140,142]. The use of immunosuppressor could be a concern in virus-derived cervical cancer patients, who may have recurrent cancer due to this treatment [143]. The first trial with successful pregnancy, delivery and live birth has been reported in 2014 by Brännström and collaborators [48]. However, several trials describe as a main risk the development of uterine intravascular thrombosis after organ transplantation [144]. Since there is a lack of published results, this option is still considered as experimental; nev-ertheless, the American Society for Reproductive Medicine is actively developing mul-tidisciplinary uterine transplant guidelines [145].”

  1. If there are any preventive methods for women who have not yet with cancer, please add them.

Answer: We appreciate the nice suggestion of the reviewer. However, we believe that discussing the preventive methods for healthy women not diagnosed with cancer may fall outside the main topic of the present review. At the present stage, the article seems to be already long and complex trying to touch upon the main topics and sections of fertility management in cancer patients at diverse stages. Therefore, by adding the healthy non-cancerous counterpart, we are afraid that the complexity of the manuscript will increase and will affect the clear message we intend to pass to the reader. We hope the reviewer will understand our decision.

Minor points

  1. Line 365: Check again “ovarian transposition is ca.”. Is it correct? 

Answer: We modified as follows (line 1001):

“ovarian transposition is approximately”

Reviewer 4 Report

Comments and Suggestions for Authors

Dear authors, I read your article with interest. Here are my suggestions:

- for endometrial cancer you have new ESGO guidelines for fertility sparing approach that you did not include

- for Ovarian cancer and borderline tumors, you have ESGO guidelines, that you did not include. You can find there specifics regarding FSS approach for early stage ovarian borderline tumors and low and high grade carcinomas. Also role of LND in germ cell tumors is debated and for BOT it is not recommended. Also,  the recurrence rates are higher in stage IC2, IC3 and grade 3 disease, although mainly in extraovarian sites and are, therefore, not clearly correlated with the fertility-sparing approach. Adequate counseling is, therefore, needed in this situation. For sBOT, FSS could be considered in selected patients.

- for cervical cancer radical hysterectomy after SHAPE trial is not the only option. Also, surgery is an option in early cancer, and definitive RTCT for others. There is also a fresh article in IJGC regarding vaginal trachelectomy (https://ijgc.bmj.com/content/34/6/799?utm_source=alert&utm_medium=email&utm_campaign=Int%20J%20Gynecol%20Cancer&utm_content=toc&utm_term=03062024)

- I advise you to use the latest FIGO staging for all the carcinomas and cite the literature

- LLETZ has affect on infertility regarding spontaneous abotions and preterm births - https://pubmed.ncbi.nlm.nih.gov/27469988/

- no mentioning of sentinel LN in cervical cancer

- no menitoning of different legislation in countries around europe regarding surrogacy after CC treatment (although you mention that later under endometrial cancer section)

- from the guidelines: Definitive surgery is recommended in cases of non-responders, inability to conceive, recurrence or disease progression (Level of evidence II, Grade A). You state that in case of recurrence a repeat treatment can be offered.

- article in literature (53) is on breast cancer

- the guidelines on ovarian cancer do not recommend FS approach for invasive EOC greater than fully staged FIGO stage I

- for borderline tumors there is different approach for sBOT and mBOT

- Table 1: I believe you should revise the table and update it. For example regarding SNB and LND in cervical cancer, then FIGO stage II is for RTCT not for surgery, etc.

- regarding ovarian transposition, ovaries can be moved also extraperitoneally

- it would be interesting to include statistics on how much is the risk of recurrence/progression affected by stimulation protocols

- impact of covid-19 was/is important but there is a question, if this topic needs the whole paragraph in this article. What is the meaning of such paragraph in post-covid era?

Author Response

Dear authors, I read your article with interest. Here are my suggestions:

Answer: Thank you for your helpful comments. Indeed, following your suggestions to whom we fully agree regarding the new ESGO guidelines in all mentioned tumor cases you reported, the recommendation of guidelines as you mentioned and the latest FIGO staging, we modify and change the parts in the text accordingly by using the new guidelines, the latest FIGO staging and your valuable comments. We also update the manuscript considering your suggestions, thus clarifying the suggested parts and improving the conceptuality and structure in the overall content of the text and the table.

- for endometrial cancer you have new ESGO guidelines for fertility sparing approach that you did not include

Answer: Thank you for your comment, we added the new ESGO guidelines in lines 597-613.

- for Ovarian cancer and borderline tumors, you have ESGO guidelines, that you did not include. You can find there specifics regarding FSS approach for early stage ovarian borderline tumors and low and high grade carcinomas. Also role of LND in germ cell tumors is debated and for BOT it is not recommended. Also,  the recurrence rates are higher in stage IC2, IC3 and grade 3 disease, although mainly in extraovarian sites and are, therefore, not clearly correlated with the fertility-sparing approach. Adequate counseling is, therefore, needed in this situation. For sBOT, FSS could be considered in selected patients.

Answer: Thank you for your recommendations. We added the latest guidelines in lines 706-728.

Regarding the role of LND and the BOT counseling importance, we fully agree with the reviewer and we modified the text in the appropriate parts as suggested.

- for cervical cancer radical hysterectomy after SHAPE trial is not the only option. Also, surgery is an option in early cancer, and definitive RTCT for others. There is also a fresh article in IJGC regarding vaginal trachelectomy (https://ijgc.bmj.com/content/34/6/799?utm_source=alert&utm_medium=email&utm_campaign=Int%20J%20Gynecol%20Cancer&utm_content=toc&utm_term=03062024)

Answer: Thank you for your recommendations. We added the new reference, as suggested (new reference 30).

- I advise you to use the latest FIGO staging for all the carcinomas and cite the literature

Answer: Thank you for your recommendation. To the best of our knowledge, we updated the main text in all three gynecological cancers according to your suggestion.

- LLETZ has affect on infertility regarding spontaneous abotions and preterm births - https://pubmed.ncbi.nlm.nih.gov/27469988/

Answer: We agree with the reviewer, and we added the new reference (25) regarding the effect of LLETZ in lines 411-412.

- no mentioning of sentinel LN in cervical cancer

Answer: We added this information in lines 381-382.

- no menitoning of different legislation in countries around europe regarding surrogacy after CC treatment (although you mention that later under endometrial cancer section)

Answer: Thank you for this comment. As far as the different legislation is a indisputable hot topic on the surrogacy following CC treatment, we decided to not include this part in our manuscript, to avoid increasing esponentiallyh the volume of the already rich text. We aim at providing a general overview of the possible approaches connected with fertility preservation and cancer patients management. We reported it exclusively in the emdometrial cancer section, due to the site of the cancer and the importance of this operative approach in patients management.

- from the guidelines: Definitive surgery is recommended in cases of non-responders, inability to conceive, recurrence or disease progression (Level of evidence II, Grade A). You state that in case of recurrence a repeat treatment can be offered.

Answer: We apologize for this mistake. We removed this part from the text.

- article in literature (53) is on breast cancer

Answer: We apologize for this mistake. We changed the wrong reference with the new reference 62:

Ko, E.M.; Walter, P.; Jackson, A.; Clark, L.; Franasiak, J.; Bolac, C.; Havrilesky, L.J.; Secord, A.A.; Moore, D.T.; Gehrig, P.A.; et al. Metformin Is Associated with Improved Survival in Endometrial Cancer. Gynecol Oncol 2014, 132, 438–442, doi:10.1016/j.ygyno.2013.11.021.

- the guidelines on ovarian cancer do not recommend FS approach for invasive EOC greater than fully staged FIGO stage I

Answer: We added this information in lines 704-705.

- for borderline tumors there is different approach for sBOT and mBOT

Answer: We agree. Indeed, we added this information in lines 759-761.

- Table 1: I believe you should revise the table and update it. For example regarding SNB and LND in cervical cancer, then FIGO stage II is for RTCT not for surgery, etc.

Answer: We thank the reviewer for this suggestion and concern. In our table we are including the fertility preservation options and applications reported in our text and closely related to reproductive/fertility potential outcome. To the best of our efforts, we searched for papers including SNB and LND that includes reproductive outcomes, but could not find papers suitable for implementing the present table. We hope the reviewer will find the present form of Table 1 suitable for our manuscript.

- regarding ovarian transposition, ovaries can be moved also extraperitoneally

Answer: We added this information with related reference in the amended text (line 994).

- it would be interesting to include statistics on how much is the risk of recurrence/progression affected by stimulation protocols

Answer: We added recurrence information with related references in the amended text (lines 1048-1053).

- impact of covid-19 was/is important but there is a question, if this topic needs the whole paragraph in this article. What is the meaning of such paragraph in post-covid era?

Answer: We thank the reviewer for this challenging criticism. As we mention at the end of our parahraph, we are reporting the experiences and challenges encountered during one of the most difficult era worldwide regarding the pandemic of Covid-19. The unexpected explotion of this viral pandemic froze and immobilized the whole world on several perspectives, among which social, medical and psychological health. During the 2 years and more of pandemic emergency state, numerous countries were completely blocked by total lockdown making impossible the normal daily management of highly sensitive patients that unfortunately could not stop oncological therapies, therefore reducing more and more their fertility window. We think that this topic deserves a small lightspot in our review to serve as a lesson to be learned from the past, emphasizing the need of important guidelines in the management of patients during future unexpected emergency statuses.

In addition, we added a small paragraph regarding the possible side effects of vaccine against Covid-19 regarding fertility potential.

Round 2

Reviewer 1 Report

Comments and Suggestions for Authors

Nothing to revise.

Reviewer 2 Report

Comments and Suggestions for Authors

In the revised manuscript, the authors have addressed most of my suggestions. 

Comments on the Quality of English Language

 I feel that the manuscript requires minor/moderate editing of English language.

Reviewer 3 Report

Comments and Suggestions for Authors

I think that the revised manuscript has been improved.

Comments on the Quality of English Language

Minor editing of English language required.

Reviewer 4 Report

Comments and Suggestions for Authors

Dear authors, thank you for the review. I have no other questions or remarks.